# The Role of Magnesium in Depression, Migraine, Alzheimer’s Disease, and Cognitive Health: A Comprehensive Review

**DOI:** 10.3390/nu17132216

**Published:** 2025-07-04

**Authors:** Péter Varga, Andrea Lehoczki, Mónika Fekete, Tamás Jarecsny, Agata Kryczyk-Poprawa, Virág Zábó, Dávid Major, Vince Fazekas-Pongor, Tamás Csípő, János Tamás Varga

**Affiliations:** 1Institute of Preventive Medicine and Public Health, Faculty of Medicine, Semmelweis University, 1085 Budapest, Hungary; varga.peter@semmelweis.hu (P.V.); ceglediandi@freemail.hu (A.L.); fekete.monika@semmelweis.hu (M.F.); zabo.virag@semmelweis.hu (V.Z.); major.david@semmelweis.hu (D.M.); pongor.vince@semmelweis.hu (V.F.-P.); csipo.tamas@semmelweis.hu (T.C.); 2Health Sciences Division, Doctoral College, Semmelweis University, 1085 Budapest, Hungary; 3Fodor Center for Prevention and Healthy Aging, Semmelweis University, 1085 Budapest, Hungary; 4Department of Neurology and Stroke, Saint John’s Central Hospital of North Buda, 1125 Budapest, Hungary; jarecsny.tamas@janoskorhaz.hu; 5Department of Inorganic Chemistry and Pharmaceutical Analytics, Faculty of Pharmacy, Jagiellonian University Medical College, 30-688 Kraków, Poland; agata.kryczyk@uj.edu.pl; 6Department of Pulmonology, Semmelweis University, 1085 Budapest, Hungary

**Keywords:** magnesium, depression, migraine, Alzheimer’s disease, cognitive health, neurotransmitters, neurodegeneration, oxidative stress, inflammation, HPA axis

## Abstract

Magnesium is an essential mineral involved in hundreds of biochemical reactions, with particular relevance to maintaining neural homeostasis, modulating neurotransmitter systems, and regulating inflammatory and oxidative stress mechanisms. This comprehensive review aims to evaluate the potential role of magnesium in the pathophysiology and treatment of three prevalent neurological and psychiatric disorders—depression, migraine, and Alzheimer’s disease—as well as its broader implications for cognitive health. Current research suggests that magnesium deficiency is associated with the development of depression, as magnesium influences glutamatergic and GABAergic neurotransmission, as well as the activity of the hypothalamic–pituitary–adrenal (HPA) axis, both of which play critical roles in stress responses and mood regulation. Additionally, magnesium’s anti-inflammatory properties may contribute to the alleviation of depressive symptoms. In the context of migraine’s pathophysiology, magnesium plays a role in regulating cerebral vascular tone, modulating the trigeminovascular system, and reducing neuronal hyperexcitability, which may explain the observed correlation between magnesium levels and the incidence of migraines. Regarding Alzheimer’s disease, preclinical and epidemiological studies suggest that magnesium may contribute to modulating neurodegenerative processes and preserving cognitive function; however, due to the heterogeneity of the current findings, further longitudinal and interventional studies are necessary to determine its precise clinical relevance. This review aims to enhance the understanding of the relationship between magnesium and these disorders through a narrative review of relevant clinical studies. The findings may provide insights into the potential therapeutic applications of magnesium and guide the future directions of the research into the prevention and treatment of depression, migraine, and Alzheimer’s disease and overall cognitive health.

## 1. Introduction

Magnesium is an essential macroelement that serves as a cofactor in more than 300 enzymatic reactions in the human body, playing a key role in nervous system function, energy production, cellular homeostasis, and the regulation of inflammatory and oxidative processes [1]. Magnesium ions are crucial for maintaining the balance of neurotransmitters such as serotonin, dopamine, and glutamate, which have direct effects on mood and cognitive functions [2]. Additionally, magnesium is involved in regulating the hypothalamic–pituitary–adrenal (HPA) axis, one of the primary systems controlling the body’s stress response [3].

In recent years, growing research interest has been directed toward investigating the potential role of magnesium in various mental and neurological disorders, particularly in depression, migraine, and Alzheimer’s disease [4,5]. Epidemiological and clinical studies suggest that magnesium deficiency may be associated with the onset and progression of these conditions, whereas adequate magnesium supplementation may exert a protective effect [5,6].

The pathophysiology of depression is multifaceted, involving inflammatory processes [7,8], impaired neuroplasticity, and dysfunction of the monoaminergic system [9]. In addition to these mechanisms, depression itself is a well-established risk factor for accelerated cognitive decline and the development of dementia, including both Alzheimer’s disease and vascular cognitive impairments [10], likely mediated through chronic inflammation, HPA axis dysregulation, and reduced hippocampal neurogenesis [7,11,12,13,14,15,16,17,18,19]. Several studies have demonstrated an association between low magnesium levels and depressive symptoms [5,20,21,22,23], as magnesium influences glutamatergic, GABAergic, and monoaminergic neurotransmission while reducing the levels of pro-inflammatory cytokines, which play a central role in depression’s pathomechanism [24,25]. Some randomized controlled trials suggest that magnesium supplementation may be an effective adjunctive therapy for depression, particularly in patients with documented magnesium deficiency [3,21,26].

Migraine’s pathogenesis involves cortical hyperexcitability, dysregulation of cerebral blood flow, and activation of the trigeminovascular system [27]. Studies indicate that migraine patients frequently exhibit low magnesium levels in their blood and cerebrospinal fluid, which may contribute to instability of cerebral vascular tone and excessive neuronal excitability [28,29]. Some clinical trials suggest that magnesium supplementation may reduce the frequency, duration, and intensity of migraine attacks, particularly in individuals with magnesium deficiency [29,30].

Alzheimer’s disease is the most common neurodegenerative disorder [31], with its pathogenesis involving oxidative stress, inflammation [32,33,34,35,36,37], microbial-infection-related processes [8,38,39,40], mitochondrial dysfunction [41,42,43], microvascular dysfunction [44], dysregulation of the cerebral blood flow [45,46], microbleeds [47], blood–brain barrier disruption [48,49], accelerated senescence [48], impaired proteostasis [50], amyloid-beta (Aβ) deposition [51], tau pathologies [52,53], and disruption of the white matter’s integrity [54] and functional connectivity [55,56,57,58,59,60]. The pathogenesis of Alzheimer’s disease is modulated by APOE status [61,62] and other genetic factors [63,64,65,66,67]; a range of cardiovascular risk factors [68,69,70,71]; and lifestyle [72,73,74,75,76,77,78,79,80,81,82] and environmental risk factors [83,84,85]. Given the global increase in dementia cases and the limited efficacy of the current pharmacologic interventions, prevention strategies targeting modifiable factors such as dietary factors warrant closer examination. Several epidemiological studies indicate that magnesium deficiency is associated with neurodegeneration, Aβ accumulation, and cognitive decline, whereas an adequate magnesium intake may exert protective effects on cognitive health [86,87,88]. However, clinical studies on Alzheimer’s disease and magnesium supplementation have yielded conflicting results, necessitating further research to clarify its therapeutic potential.

This comprehensive review aims to systematically evaluate the role of magnesium in the pathogenesis and treatment of depression, migraine, and Alzheimer’s disease and in cognitive health. To our knowledge, this is the first narrative review to compare the effects of magnesium across these three major neurological disorders, with a particular focus on their shared pathophysiological mechanisms. Special attention is given to the extent to which available clinical data support the efficacy of magnesium supplementation in these conditions. Its findings may contribute to elucidating the potential clinical applications of magnesium and identifying future research directions.

## 2. Methods

A comprehensive literature search was conducted using four major scientific databases: PubMed, Scopus, Web of Science, and Embase. Relevant studies published in both English and Hungarian were considered. The search covered the period from 2000 to 2025, reflecting the substantial advancements in neuroscience and psychiatry related to magnesium research over the past two decades.

The search strategy employed a combination of keywords and Boolean operators to identify relevant publications. Key terms included magnesium, Mg, depression, major depressive disorder, migraine, headache disorders, Alzheimer’s disease, cognitive decline, dementia, cognitive health, neurotransmitters, glutamate, GABA, serotonin, dopamine, inflammation, pro-inflammatory cytokines, IL-6, TNF-alpha, oxidative stress, ROS, antioxidants, hypothalamic–pituitary–adrenal axis, and cortisol. To optimize the search’s efficiency, Boolean operators were applied as follows:AND was used to link related concepts (e.g., magnesium AND depression AND neurotransmitters);OR connected synonyms (e.g., depression OR major depressive disorder);NOT was used to exclude irrelevant studies.

### 2.1. Application of the PICO Model

To clearly formulate the research question and ensure focused data extraction, the PICO model (Population, Intervention, Comparison, Outcome) was applied, as summarized in Table 1.

### 2.2. The Inclusion and Exclusion Criteria

The eligibility of studies was determined using predefined inclusion and exclusion criteria to ensure the relevance and methodological rigor of the selected literature (Table 2).

### 2.3. The Data Extraction and Analysis Methods

The study selection was performed by two independent researchers, and any discrepancies were resolved through consensus-based discussion. The extracted data included the following variables:Study type (e.g., RCT, longitudinal study);Sample size and demographic characteristics;Magnesium dosage and administration route;Measurement methods and primary outcome variables.

Our findings aim to contribute to a better understanding of the therapeutic role of magnesium in depression, migraine, and Alzheimer’s disease. This review article includes a total of 43 clinically relevant studies, organized into tables, comprising 219,224 participants. The study selection process is illustrated in Figure 1.

In the following sections, we examine the potential role of magnesium in three major neurological and psychiatric conditions—depression, migraine, and Alzheimer’s disease—using a consistent structure that includes a brief overview of their pathophysiology, the current evidence, the limitations of the literature, and possible clinical applications.

## 3. The Role of Magnesium Supplementation in the Treatment of Depression

In recent years, an increasing body of scientific evidence has supported a close relationship between depression and low magnesium levels [3,5,89,90,91,92,93,94,95]. Several epidemiological and clinical studies have confirmed that depressed patients often exhibit low magnesium levels, which may contribute to the onset or worsening of the disease [6,89,90,91]. Magnesium plays a key role in regulating neurological processes, including the balance of neurotransmitters (such as glutamate, GABA, and serotonin), as well as reducing oxidative stress and inflammation [2]. Nevertheless, the effectiveness of magnesium supplementation in the treatment of depression has been evaluated in only a limited number of studies, and the results remain inconclusive.

Various randomized clinical trials have shown mixed results regarding the association of magnesium supplementation and depression, but several studies have indicated an improvement in serum magnesium levels and depressive symptoms (Table 3). Afsharfar et al. [96] conducted a study in which 46 depressed patients received 500 mg of magnesium oxide daily for 8 weeks as monotherapy. In the treatment group, there was a significant improvement in their Beck Depression Inventory (BDI) scores (*p* = 0.01), and their serum magnesium levels increased (*p* = 0.001), although there was no significant change in their Brain-Derived Neurotrophic Factor (BDNF) levels. Barragán-Rodríguez et al. [97] examined 23 elderly patients with type 2 diabetes and hypomagnesemia. Although no significant difference in depressive symptoms was found between the magnesium chloride and Imipramine groups (*p* = 0.27), the magnesium chloride group showed significantly higher serum magnesium levels (*p* < 0.0005).

Rodríguez-Morán et al. [98] conducted a study involving 60 long COVID patients who received 1300 mg of magnesium chloride and 4000 IU of vitamin D daily for 4 months. In the intervention group, a significant decrease in their BDI scores was observed (*p* < 0.01), while the control group also showed improvements, albeit to a lesser extent (*p* < 0.05). Rajizadeh et al. [99] conducted a study with 60 depressed patients who received magnesium oxide supplementation for 8 weeks, resulting in a significant improvement in both their BDI-II scores (*p* = 0.02) and serum magnesium levels (*p* = 0.002). The magnesium group showed a greater reduction in depressive symptoms compared to that in the placebo group.

Abiri et al. [100] studied 108 obese women who received vitamin D and magnesium. The results showed a significant improvement in the depressive symptoms and inflammatory markers in the intervention group (vitamin D and magnesium) compared to that under vitamin D supplementation only, but no significant difference was observed between the different treatment groups regarding the severity of depression. In the study by Shakya et al. [101], the relationship between dietary habits and depression was investigated. The results indicated that a “prudent” diet (a generally healthy, plant-forward diet) was inversely related to depressive symptoms, while a “Western” diet contributed to their development. Fard et al. [102] studied 99 women postpartum who were given either magnesium or zinc sulfate, but no significant improvement in their depression levels was found.

Mehdi et al. [103] intravenously administered magnesium sulfate (4 g in 5% dextrose solution) over 8 days, followed by a 5-day washout period. The participants (*n* = 12) had mild to moderate treatment-resistant depression. The intervention led to a significant increase in their serum magnesium levels (*p* = 0.02) and a significant reduction in their PHQ-9 depression scores (*p* = 0.02). Tarleton et al. [23] evaluated the effects of daily oral magnesium supplementation (248 mg) over 6 weeks in 126 adults. Significant improvements were observed in both their depressive and anxiety symptoms (PHQ-9: –6.0 points; GAD-7: –4.5 points), with effects emerging as early as 2 weeks.

Ryszewska-Pokraśniewicz et al. [22] studied 37 depressed patients who received fluoxetine and magnesium. No significant differences were observed on the Hamilton Depression Rating Scale (HDRS), but magnesium treatment increased the likelihood of effectiveness, particularly in patients with lower baseline HDRS scores. In the study by Pouteau et al. [104], 264 healthy adults were tested, and those who received both magnesium and vitamin B6 showed a 24% greater improvement in their DASS-42 stress scores compared to these values in those who only took magnesium, particularly in those with high stress levels (*p* = 0.02). Finally, Derom et al. [105] and Nazarinasab et al. [106] found no significant long-term association between magnesium intake and the risk of depression, although smaller improvements were observed in some groups.

Overall, these studies suggest that magnesium supplementation may have a beneficial effect on depressive symptoms, but the degree and consistency of its effect vary. While several studies have found significant improvements, others were less successful. Further research is necessary to determine the optimal dosage, underlying mechanisms, and long-term outcomes better. These contradictory results may be attributed to several factors. Different studies used varying sample sizes, measurement methods, and magnesium supplementation protocols, making it difficult to compare the results and draw general conclusions. Additionally, depression is a complex, multifactorial condition that involves not only nutritional factors but also genetic, environmental, and psychological elements. Based on the existing evidence, the relationship between magnesium and depression appears clear, but the therapeutic effect of magnesium supplementation requires further large-scale, well-designed clinical trials. The following table summarizes the clinical outcomes of randomized trials investigating magnesium supplementation in depression (Table 3).

## 4. The Role of Magnesium in the Pathophysiology of Migraine

Magnesium plays an essential role in neurological function, including neurotransmission, regulation of vascular tone, and inhibition of the NMDA receptors [6]. Several studies have demonstrated that patients with migraine frequently exhibit low serum and intracellular magnesium levels, particularly during migraine attacks [28,29,30,107,108]. Magnesium deficiency may contribute to neuronal hyperexcitability, the development of cortical spreading depression (CSD), and enhanced activation of the trigeminovascular system—all of which are central mechanisms in the pathogenesis of migraine attacks [109]. Oral magnesium used for prophylactic purposes (e.g., magnesium oxide at a daily dose of 400–600 mg) has been shown to be effective in several randomized controlled trials in reducing the frequency and intensity of migraine attacks, especially in cases of menstrual migraine and migraine with aura [30,110].

Numerous clinical studies have investigated the use of intravenous magnesium sulfate in the treatment of migraine attacks, yielding partly conflicting results [111,112]. Cete et al. [113] evaluated 113 adult migraine patients treated with MgSO_4_ (2 g) or metoclopramide (10 mg). Both groups demonstrated significant improvements on the Visual Analogue Scale (VAS) exceeding 25 mm within 30 min; however, no statistically significant difference was observed between the groups. The placebo group required more rescue medication.

Corbo et al. [114] compared magnesium sulfate combined with metoclopramide and metoclopramide with a placebo in acute migraine patients and found that the magnesium group showed lower pain reductions and that fewer patients achieved normal functioning. Conversely, Demirkaya et al. [115] reported that a single 1 g MgSO_4_ infusion led to an 87% pain-free status and the complete resolution of symptoms in 100% of patients, with only mild and non-serious side effects; however, it should be noted that this was a pilot study with a small sample size.

Bigal et al. [116] demonstrated that magnesium sulfate was particularly effective in reducing pain and symptoms in migraine with aura, while in migraine without aura, only photophobia and phonophobia showed significant improvements. Shahrami et al. [117] also observed faster and more pronounced pain relief with MgSO_4_ compared to that with dexamethasone plus metoclopramide.

Magnesium’s efficacy as an adjunct therapy is supported by Matin et al. [118], who found significant reductions in migraine-associated inflammatory markers and symptoms when vitamin B_12_ and high-intensity interval training were combined with magnesium supplementation. Other studies, such as Kandil et al. [119] and Gaul et al. [120], showed comparable analgesic effects between magnesium, metoclopramide, and prochlorperazine, as well as modest benefits from magnesium-containing combination supplements on migraine days.

In emergency settings, Rahimdel et al. [121] confirmed the effectiveness of MgSO_4_ in reducing pain intensity 60 and 90 min post-administration. However, Ginder et al. [122] highlighted that magnesium’s efficacy was not correlated with serum magnesium levels. In prophylactic use, Khani et al. [123] reported that three months of treatment with magnesium, sodium valproate, or their combination significantly reduced the frequency and severity of migraines, with the combination therapy showing superior outcomes. Similarly, Karimi et al. [124] found comparable efficacy between magnesium oxide and sodium valproate in an 8-week crossover trial. Tarighat Esfanjani et al. [125] investigated magnesium oxide, L-carnitine, and their combination over 12 weeks, revealing significant improvements in all intervention groups regarding the frequency and severity of migraines. Köseoglu et al. [126] demonstrated that three months of oral magnesium citrate treatment significantly decreased the frequency and severity of migraines and improved the cortical blood flow in patients with migraine without aura. Overall, the literature suggests that various forms of magnesium can be effective in both acute migraine management and prophylaxis, particularly in migraine with aura. Magnesium treatments generally have a favorable safety profile, although further research is warranted to understand their mechanisms better and optimize the therapeutic protocols. A summary of key clinical trials evaluating magnesium in both acute and prophylactic treatment of migraines is presented in Table 4.

## 5. The Role of Magnesium in Dementia Prevention and Slowing the Progression of Alzheimer’s Disease

Dementia is one of the leading causes of disability and mortality among older adults worldwide [127]. Due to the aging global population, both the number of affected individuals and the associated healthcare costs are increasing exponentially [128]. According to data from 2023, over 55 million people are currently living with dementia, and this number is projected to rise to 139 million by 2050 [129,130]. In light of this, there is an urgent need to develop effective, evidence-based preventive strategies.

Alzheimer’s disease is the most prevalent form of neurodegenerative dementia and poses a growing global public health challenge [131]. It is characterized by progressive neuronal loss affecting multiple brain regions. Despite extensive research efforts, no effective cure currently exists [132]. The limited success of pharmacological interventions over recent decades has shifted the focus toward prevention and the identification of modifiable risk factors involved in disease pathogenesis [133]. In this context, magnesium has garnered increasing scientific interest due to its involvement in numerous neurobiological processes potentially influencing the onset and progression of AD [134]. Although the precise mechanisms remain incompletely elucidated, recent studies have identified several pathways through which magnesium may exert a protective effect [135].

Magnesium plays a critical role in maintaining synaptic plasticity and neuronal function. It modulates the activity of the N-methyl-D-aspartate glutamate receptors, which are essential for learning and memory [2]. Excessive activation of these receptors can lead to excitotoxicity, a process that contributes to neuronal damage and is implicated in AD progression [136]. By attenuating NMDA receptor overactivation, magnesium may confer neuroprotective benefits [137]. Furthermore, magnesium exhibits antioxidant properties that may mitigate oxidative stress—an established contributor to AD pathophysiology [2]. Oxidative stress arises from the excessive generation of reactive oxygen and nitrogen species, which damage the neuronal structures over time [138]. Adequate magnesium levels may support endogenous antioxidant defenses and help counteract these harmful effects [139].

Chronic neuroinflammation is another major factor in AD development [140]. Activated microglia and the overproduction of pro-inflammatory cytokines contribute to sustained neuroinflammatory responses and subsequent neurodegeneration [141]. Magnesium may exert immunomodulatory effects that reduce central nervous system inflammation and slow the associated neuronal damage [135]. Magnesium is also involved in the regulation of two hallmark pathologies of AD: amyloid-beta plaques and neurofibrillary tangles composed of hyperphosphorylated tau protein [142]. Emerging evidence suggests that magnesium may influence Aβ peptide aggregation and clearance, as well as attenuating pathological tau phosphorylation—mechanisms that could lessen the morphological burden of AD [143,144]. Additionally, magnesium is indispensable to mitochondrial function and cellular energy metabolism [145]. Mitochondrial dysfunction is a fundamental component of AD pathophysiology that contributes to neuronal energy failure and apoptosis [146]. By preserving mitochondrial integrity, magnesium may help maintain neuronal viability and slow disease progression [147].

## 6. Epidemiological and Clinical Evidence

### 6.1. Serum Magnesium Levels and Dementia Risk

Numerous observational and interventional studies have investigated the extent to which magnesium deficiency (hypomagnesemia) may contribute to the development and progression of Alzheimer’s disease [4,6,21,148,149,150]. A randomized controlled trial reported that the calcium-to-magnesium ratio may influence the methylation patterns of the *APOE* gene, which plays a key role in the pathomechanism of AD [151]. The authors observed that modified magnesium intake could affect cognitive function through epigenetic mechanisms, thereby potentially influencing the risk of AD [151]. Further support for this association comes from a comprehensive meta-analysis that summarized data from 21 studies [152]. The analysis found that individuals with AD had significantly lower serum and plasma magnesium levels compared to those in healthy controls (mean difference: –0.09 mmol/L, *p* < 0.001), suggesting that magnesium deficiency may be linked to the prevalence of AD [152].

The REGARDS study, a large, long-term American cohort, concluded that low serum magnesium levels (<0.75 mmol/L) are associated with an increased risk of cognitive decline [153]. Magnesium concentrations in the upper–normal range (0.85–0.89 mmol/L) were associated with a protective effect, whereas levels exceeding the normal range did not confer additional benefits. These findings suggest that maintaining adequate—but not excessive—magnesium levels may be linked to preserved cognitive health in older adults [153]. Another longitudinal study demonstrated that both low (≤0.79 mmol/L) and high (≥0.90 mmol/L) serum magnesium levels were associated with an increased overall risk of dementia, particularly non-Alzheimer’s subtypes such as vascular dementia. The lowest risk was observed within the middle quintile (~0.85 mmol/L), which is considered the optimal range. The mediation analysis from the same study indicated that diabetes may be a major mediating factor in the relationship between low magnesium levels and dementia risk, whereas smoking, stroke, and hypertension appeared to play more minor roles [154]. The Atherosclerosis Risk in Communities (ARIC) study also demonstrated that low midlife serum magnesium levels were associated with a 24% increased risk of developing dementia later in life, compared to higher magnesium levels, used as a reference. Although the incidence of dementia was elevated in the low-magnesium group, no significant differences were found in the rates of cognitive decline (e.g., memory and executive functions), suggesting that magnesium levels may influence the risk of dementia onset rather than the speed of cognitive deterioration [155]. Finally, the findings from the Rotterdam Study indicated that both low (≤0.79 mmol/L) and high (≥0.90 mmol/L) serum magnesium levels were associated with a 30% increased risk of dementia compared to that for the middle reference range (0.80–0.89 mmol/L). These results imply that both extremes of magnesium status may be detrimental to cognitive health and that maintaining an optimal, balanced magnesium level may have preventive importance [156]. Based on the epidemiological evidence, maintaining an adequate magnesium intake—within a recommended dietary allowance of 400–420 mg/day for men and 310–320 mg/day for women—may play a particularly important role in the prevention of dementia, especially in older adults and individuals with metabolic risk factors such as diabetes or hypertension. A growing body of scientific research supports the notion that magnesium is a critical modulator of neurodegenerative processes and may contribute to the prevention or slowing of Alzheimer’s disease progression [135,145,157].

### 6.2. Magnesium Intake and Supplementation and Cognitive Outcomes

Numerous studies have demonstrated the role of magnesium in regulating cognitive functions and dementia risk [4,6,148,158]. Zhu et al. [151], in a cohort of 250 participants—primarily individuals over 65 years—showed that reducing the dietary calcium-to-magnesium ratio to approximately 2.3 through personalized magnesium supplementation resulted in a 9.1% improvement in cognitive function, which was partly mediated by epigenetic changes, particularly alterations in APOE gene methylation patterns (*p* = 0.03). This improvement was modest but potentially meaningful for early cognitive decline. In contrast, Ni et al. [159], during a 12-week intervention using magnesium L-threonate, found no significant improvement in chronic pain, psychological symptoms, or cognitive performance among breast cancer surgery patients, suggesting the need for more complex therapeutic approaches.

Long-term cohort studies (Ozawa et al., [160]; Cherbuin et al., [161]) support the association between a higher magnesium intake and a reduced dementia risk, especially for vascular dementia. Lo et al. [162] described a non-linear relationship in postmenopausal women, where a moderate magnesium intake was associated with a decreased incidence of mild cognitive impairment and dementia, whereas the lowest and highest quintiles showed no protective effect. Kimura et al. [163] linked higher consumption of vegetables—and the associated intake of magnesium, calcium, potassium, and vitamins—to reduced risks of dementia and Alzheimer’s disease. Similarly, Tao et al. [164] found that a higher total magnesium intake (from the diet and supplements) was associated with a better global cognitive performance in older adults from the U.S., particularly among women, non-Hispanic Whites, and those with sufficient serum vitamin D levels. In a separate 6-year follow-up study, Tao et al. [165] reported that a higher dietary magnesium intake was linked to a lower risk of cognitive impairments in men and to a lesser extent in women, regardless of vitamin D status.

Luo et al. [166] emphasized the importance of the calcium-to-magnesium ratio, reporting that a low dietary calcium and magnesium intake, and a Ca:Mg ratio ≤1.69 combined with a high magnesium intake (>267.5 mg/day), increased dementia risk (HR = 3.97). Tzeng et al. [167], in a 10-year study, demonstrated that magnesium oxide supplementation significantly lowered the incidence of dementia (HR = 0.517; *p* = 0.001).

Zhang et al. [168] conducted a 30-day intervention in healthy adults using Magtein^®^ PS (400 mg of magnesium L-threonate and 50 mg of phosphatidylserine and vitamins C, D, and B6), which yielded significant improvements across all memory domains, particularly in older participants. Data from the UK Biobank (Takeuchi et al., [169]) revealed a non-linear relationship between nutrient intake—including magnesium—and dementia risk: both a very low and an excessively high magnesium intake, alongside certain lifestyle factors, increased this risk, while a moderate, balanced nutrient intake was most protective. In summary, magnesium intake plays a critical role in maintaining cognitive function and preventing dementia, particularly in older adults. Achieving the optimal effects requires maintaining a balanced calcium-to-magnesium ratio, as both insufficient and excessive magnesium intake may elevate dementia risk. Alterations in the Ca^2+^:Mg^2+^ ratio may contribute to cognitive decline by disrupting calcium signaling, which affects neuronal excitability, gene expression (including APOE methylation), and energy metabolism [151]. Lowering this ratio through an increased magnesium intake may help restore neurophysiological balance and support cognitive function. Magnesium L-threonate formulations (e.g., Magtein^®^ PS) show promising results in improving memory functions, especially among healthy elderly populations [168]. However, in certain groups—such as patients with chronic pain or psychological symptoms—magnesium supplementation alone may be insufficient, necessitating more comprehensive therapeutic strategies. Long-term prospective studies corroborate magnesium’s neuroprotective role, while the interplay of nutrients and lifestyle factors is also crucial to reducing dementia risk. Table 5 summarizes key studies investigating the role of magnesium and dietary factors in dementia risk and cognitive decline.

## 7. Prevention, Therapy, and Neurological Disorders: A Scientific Discussion on the Role of Magnesium

The current scientific evidence increasingly supports the view that magnesium plays a key role in the pathophysiology, prevention, and potential treatment of several neurological and psychiatric conditions [135]. A growing body of literature links magnesium status to disorders such as depression, migraine, and dementia and various forms of neurodegeneration [4,6,118,148]. Both epidemiological and experimental data suggest that magnesium deficiency—either as a cause or a consequence—may significantly contribute to the onset and progression of these diseases [6,148].

In the case of depression, multiple observational studies have demonstrated a correlation between low serum magnesium levels and depressive symptoms [21,22,23]. Magnesium is known to regulate key neurotransmitters such as serotonin and GABA and also modulate the HPA axis and NMDA receptor activity—mechanisms that are dysregulated in depression [172]. Interventional studies using magnesium supplementation, particularly with magnesium glycinate or magnesium L-threonate, have shown symptom improvements in mild to moderate depression, though the results remain mixed due to variations in the sample sizes, dosages, and treatment durations [173]. Despite this heterogeneity, magnesium is increasingly viewed as a potential adjunctive therapy, especially in patients with documented magnesium deficiency or a poor response to standard antidepressants [3].

In migraine prophylaxis, oral magnesium supplementation—often in the form of magnesium citrate or magnesium oxide—has demonstrated efficacy in reducing the frequency and severity of attacks, particularly among patients experiencing migraine with aura [29]. Magnesium appears to inhibit cortical spreading depression and stabilize vascular tone, which are both implicated in migraine’s pathophysiology [109]. However, studies on intravenous magnesium sulfate supplementation for acute migraine attacks have yielded conflicting results [30,111]. While some randomized trials reported rapid pain relief following infusion, others found no significant benefit compared to that under a placebo or the conventional treatments [111,113,115]. These discrepancies may be attributable to differences in magnesium’s bioavailability, the timing of administration, or underlying migraine subtypes.

In the context of dementia, especially Alzheimer’s disease, magnesium is emerging as a key player in cognitive health and neuroprotection [6,148]. Epidemiological studies have identified a U-shaped relationship between serum magnesium levels and dementia risk: both hypomagnesemia and hypermagnesemia appear to increase the likelihood of cognitive decline [156]. The optimal serum range associated with a lower dementia risk is estimated at 0.80–0.89 mmol/L [150]. Preclinical studies further show that magnesium supplementation can improve synaptic plasticity and dendritic spine morphology and reduce tau hyperphosphorylation, one of the hallmarks of Alzheimer’s pathology [135]. While magnesium has shown potential neuroprotective effects, several studies have reported no significant cognitive benefit, particularly in the advanced stages of Alzheimer’s disease [169].

Intracellular magnesium exerts its neuroprotective effects through multiple mechanisms: it modulates calcium homeostasis, reduces oxidative stress, attenuates neuroinflammation, and enhances the ATP production in the mitochondria [135]. For example, in animal models, magnesium L-threonate increased brain magnesium levels and reversed cognitive deficits by promoting synaptic density [174]. Moreover, magnesium supplementation has been shown to inhibit the activation of the NF-κB pathway, thereby reducing pro-inflammatory cytokine production, which is known to contribute to neurodegenerative progression [175].

Despite compelling preclinical evidence, translating magnesium’s neuroprotective effects into clinical practice is complex [135]. Factors such as blood–brain barrier permeability, magnesium’s bioavailability, and individual metabolic differences can influence the therapeutic outcomes [4]. Furthermore, the form of magnesium, its dosage, and the treatment duration are not standardized across studies, making cross-comparisons difficult. Different magnesium salts have varying bioavailability, which can affect both its absorption and the therapeutic outcomes. Patients with acute neurological conditions (e.g., stroke, traumatic brain injury) may not benefit substantially from magnesium therapy due to its slow cellular uptake and the narrow therapeutic window in such emergencies [176,177]. In contrast, chronic neurodegenerative disorders present a more favorable context for sustained magnesium intervention [6]. There is also emerging interest in magnesium’s indirect cognitive benefits, such as through improved sleep regulation and anxiolytic effects, both of which influence long-term cognitive resilience [178]. Additionally, magnesium may modulate nitric oxide synthesis, thereby improving cerebral perfusion, another critical factor in cognitive preservation [135].

Magnesium represents a promising, multifaceted agent in the context of neurological and psychiatric disorders [145]. Its role as a neuroprotective nutrient is supported by experimental and epidemiological findings, although robust, long-term clinical trials are needed to determine its therapeutic potential across specific patient populations. While not yet established as a first-line treatment, magnesium supplementation may hold clinical utility as an adjunctive strategy, particularly in individuals with a suboptimal magnesium status or a heightened neuroinflammatory burden [135].

## 8. Recommendations and Future Directions

To clarify the therapeutic potential of magnesium in neurological disorders further, future research should prioritize the implementation of standardized protocols, ensuring consistency in the dosages, treatment durations, and outcome measures. Long-term follow-up studies are essential to evaluate its sustained efficacy and safety across different clinical scenarios. Comparative analyses across diverse populations would also provide valuable insights into demographic or genetic factors that may influence magnesium metabolism and the treatment response. Additionally, a deeper exploration of the intracellular mechanisms of magnesium’s action is warranted to understand its role at the cellular and molecular levels better. Investigating the potential for personalized magnesium supplementation—based on genetic and epigenetic markers—may offer a path toward optimizing its therapeutic efficacy and tailoring interventions to individual patient profiles.

## 9. Limitations

When interpreting the available data on magnesium’s role in neurological disorders, several critical limitations must be acknowledged. First, there is considerable heterogeneity among studies in terms of the methodology, the magnesium dosage, the duration of the intervention, and participant characteristics (e.g., age, comorbidities). The magnesium dosages and administrative routes also varied greatly across the included studies, which may have affected the bioavailability of magnesium. This diversity complicates direct comparisons and limits the generalizability of the findings. Moreover, many studies have involved small sample sizes and relatively short follow-up periods, raising concerns about the robustness and long-term sustainability of the observed effects. The presence of comorbidities and the frequent use of combined therapeutic approaches—such as concurrent pharmacological treatments, vitamin supplementation, or lifestyle modifications—confound the ability to isolate the specific impact of magnesium further. Another methodological concern is the predominant reliance on serum magnesium levels, which may not accurately reflect magnesium status at the intracellular level, where many of its physiological actions occur. Furthermore, the lack of consistent reporting of pre-treatment serum magnesium levels in many clinical trials makes it difficult to determine whether therapeutic effects are dependent on baseline deficiency. Lastly, many of the outcomes, particularly those related to psychological symptoms, were assessed using self-reported questionnaires. These tools are inherently susceptible to placebo effects and subjective biases, which may have influenced the perceived efficacy of magnesium supplementation.

## 10. Conclusions

In summary, magnesium plays a significant role in both the prevention and adjunctive management of neurological disorders. Based on the current body of evidence, magnesium supplementation may offer a promising therapeutic option in the treatment of depression, particularly in mild to moderate cases. However, further well-designed, large-scale, and long-term clinical trials are required to refine the clinical recommendations and determine the precise conditions under which magnesium exerts its beneficial effects. In the context of migraine, magnesium—especially when used prophylactically—has demonstrated efficacy and safety, notably in patients experiencing aura. Nonetheless, its role in acute treatment remains less clearly defined, as intravenous administration has yielded mixed results across studies. Regarding dementia and Alzheimer’s disease, maintaining adequate magnesium levels appears to be of preventive value, especially in older adults and individuals with metabolic risk factors. The optimal magnesium status may contribute to a reduction in neurodegenerative processes due to its antioxidant, anti-inflammatory, and neuroprotective mechanisms.

## Figures and Tables

**Figure 1 nutrients-17-02216-f001:**
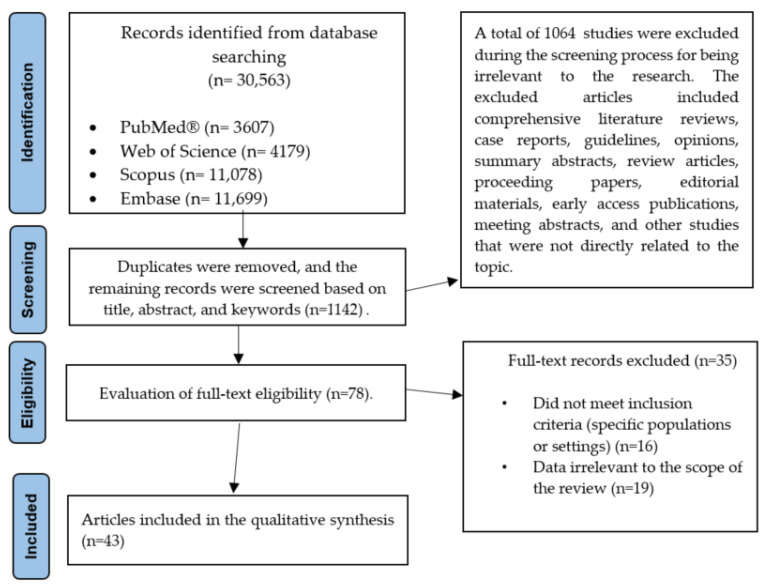
A flowchart illustrating the selection process for the included articles.

**Table 1 nutrients-17-02216-t001:** The application of the PICO model to defining the research question.

Category	Description
Population (P)	Adults and elderly individuals (≥18 years), patients with depression, migraine, or Alzheimer’s disease
Intervention (I)	Magnesium supplementation in any form (oral, intravenous), the effects of a magnesium-rich diet
Comparison (C)	Placebo or other standard treatments, comparison between individuals with low and normal magnesium levels
Outcome (O)	Improvement in mood and cognitive function, reduction in depressive symptoms (e.g., Beck Depression Inventory scores), changes in migraine attack frequency and intensity, slowing of Alzheimer’s disease progression (e.g., Mini-Mental State Examination (MMSE) score changes)

**Table 2 nutrients-17-02216-t002:** The inclusion and exclusion criteria for the reviewed studies.

Inclusion Criteria	Exclusion Criteria
Randomized controlled/clinical trials	Animal studies and in vitro research
Longitudinal and prospective cohort studies	Non-peer-reviewed articles, conference abstracts, opinion papers
Full-text, peer-reviewed scientific articles	Case reports
Studies conducted on human participants	Studies published in languages other than English or Hungarian

**Table 3 nutrients-17-02216-t003:** Magnesium and depression: clinical outcomes from randomized trials.

Study (Ref.)	Intervention	Duration	Participants	Outcomes Measured	Key Findings
Afsharfar et al. [96]	Oral magnesium oxide	500 mg/day for 8 weeks	46 depressed patients (randomized into Mg and placebo groups)	Beck Depression Inventory scores, serum BDNF, serum Mg	Significant improvement in BDI scores (*p* = 0.01) and serum Mg levels (*p* = 0.001); no change in BDNF (*p* = 0.507)
Barragán-Rodríguez et al. [97]	Oral magnesium chloride (MgCl_2_)	450 mg of elemental Mg/day for 12 weeks	23 elderly patients with type 2 diabetes and hypomagnesemia	Yasavage and Brink depression scores, serum Mg levels	No difference in depression (*p* = 0.27); higher serum Mg in the MgCl_2_ group (*p* < 0.0005)
Rodríguez-Morán et al. [98]	Oral magnesium chloride + vitamin D	1300 mg of MgCl_2_ + 4000 IU of vitamin D daily for 4 months	60 long COVID patients with mild to moderate depression	Beck Depression Inventory (BDI) scores	A significant reduction in BDI scores (*p* < 0.01 intervention; *p* < 0.05 control)
Rajizadeh et al. [99]	Oral magnesium oxide (MgO)	500 mg/day for 8 weeks	60 depressed patients (Mg vs. placebo groups)	Beck Depression Inventory-II scores, serum Mg levels	Significant reductions in BDI scores (*p* = 0.02) and Mg (*p* = 0.002); 15.65 points (Mg) vs. 10.40 (placebo)
Abiri et al. [100]	Oral magnesium + vitamin D	50,000 IU of vitamin D weekly + 250 mg of Mg daily for 8 weeks	108 obese women with mild to moderate depression	BDI-II scores, inflammatory markers, 25(OH)-D, serum Mg	Significant improvements in markers; no significant difference in BDI-II scores
Shakya et al. [101]	Dietary patterns (PCA, RRR, PLS)	N/A	1743 adults from the North West Adelaide Health Study (NWAHS)	CES-D scores, dietary patterns	‘Prudent’ diet inversely and ‘Western’ diet positively associated with depression (PCA: OR = 0.57, 2.04)
Fard et al. [102]	Oral magnesium sulfate, zinc sulfate	27 mg of zinc sulfate or 320 mg of magnesium sulfate for 8 weeks	99 women postpartum (randomized into groups)	Edinburgh Postnatal Depression Scale scores, Anxiety Inventory scores	No significant difference in depression (*p* = 0.553) or anxiety
Mehdi et al. [103]	IV magnesium sulfate	4 g of magnesium sulfate in 5% dextrose for 8 days (5-day washout)	12 subjects with mild to moderate treatment-resistant depression	Serum Mg, PHQ-9 scores	A significant increase in serum Mg (*p* = 0.02); a decrease in PHQ-9 scores (*p* = 0.02)
Tarleton et al. [23]	Oral magnesium chloride	248 mg/day for 6 weeks	126 adults with mild to moderate depression	PHQ-9 scores, GAD-7 scores, adherence, adverse effects	Significant improvements in PHQ-9 (−6.0 points, *p* < 0.001) and GAD-7 (−4.5 points, *p* < 0.001) scores
Ryszewska-Pokraśniewicz et al. [22]	Oral magnesium aspartate	120 mg/day for 8 weeks with fluoxetine	37 patients with recurrent depression disorder	HDRS scores, serum Mg levels, pharmaco-EEG	No significant changes in HDRS scores but increased effectiveness with magnesium augmentation
Pouteau et al. [104]	Oral magnesium + vitamin B6	300 mg of Mg + 30 mg of vitamin B6 for 8 weeks	264 healthy adults with stress (DASS-42 > 18)	DASS-42 stress subscale scores	A 24% greater improvement in severe stress with Mg + B6 (*p* = 0.0203)
Derom et al. [105]	Dietary magnesium intake	Median follow-up of 6.3 years	12,939 Spanish university graduates	Depression incidence (self-reported, antidepressant use)	No association between Mg intake and depression risk (*p*-trend = 0.59)
Nazarinasab et al. [106]	Oral magnesium supplement, 250 mg/day	6 weeks of Mg vs. placebo + SSRI treatment	60 patients with major depressive disorder (MDD)	Beck Depression Inventory-II scores	A significant improvement in BDI scores at 4 and 6 weeks (*p* = 0.02, *p* = 0.001)

Abbreviations: Mg: Magnesium; MgCl_2_: Magnesium Chloride; BDI: Beck Depression Inventory; BDNF: Brain-Derived Neurotrophic Factor; PCA: Principal Component Analysis; RRR: Reduced Rank Regression; PLS: Partial Least Squares; CES-D: Center for Epidemiologic Studies Depression Scale; DASS-42: Depression Anxiety Stress Scale–42 items; GAD-7: Generalized Anxiety Disorder 7-item scale; PHQ-9: Patient Health Questionnaire-9; SSRI: Selective Serotonin Reuptake Inhibitor; 25(OH)-D: 25-Hydroxyvitamin D; HDRS: Hamilton Depression Rating Scale; OR: Odds Ratio.

**Table 4 nutrients-17-02216-t004:** Summary of clinical studies on magnesium in acute and prophylactic migraine treatment.

Study (Ref.)	Intervention	Duration	Participants	Outcomes Measured	Key Findings
Magnesium supplementation only					
Demirkaya et al. [115]	IV MgSO_4_ (1 g)	Single 15 min infusion	30 patients (15 Mg, 15 placebo) with migraine	Pain, symptoms, side effects at 0 and 30 min and 2 h	87% pain-free with Mg vs. 0% with the placebo; total symptom relief: 100% vs. 20%; mild side effects
Bigal et al. [116]	IV MgSO_4_ (1 g)	Single 10 mL infusion	120 patients (60 with aura, 60 without aura)	Pain, nausea, photophobia, phonophobia, aura	With aura: significant relief (NNT = 2.7); without aura: no pain/nausea relief (NNT = 5.98); ↓ photo/phonophobia
Combined supplementation					
Cete et al. [113]	IV MgSO_4_ (2 g) + metoclopramide (10 mg)	Single 10 min infusion	113 adults with migraine (IHS criteria), three groups	VAS scores at 0, 15, and 30 min; rescue meds; recurrence at 24 h	All groups improved by >25 mm; no VAS differences; the placebo needed more rescue meds
Corbo et al. [114]	IV MgSO_4_ (2 g) + metoclopramide (20 mg) vs. placebo	Max of three doses at 15 min intervals	44 adults with acute migraine	VAS scores at 0–45 min; function; side effects	The Mg group was less effective (–16 mm); NNH = 4; worse functional outcomes
Shahrami et al. [117]	IV MgSO_4_ (1 g) vs. 8 mg dexamethasone + 10 mg metoclopramide	Single IV dose	70 adults, randomized into two equal groups	NRS scores at the baseline, 20 min, 1 h, and 2 h	MgSO_4_ led to faster and greater pain reductions (2 h NRS scores: 1.3 vs. 2.5); *p* < 0.0001
Matin et al. [118]	Oral magnesium (250 mg) + vitamin B12 (1 mg) ± HIIT	2 months	60 women, four randomized groups	CGRP levels, MIDAS scores, frequency, intensity, duration	HIIT + Mg reduced CGRP levels and migraine indicators the most; supported by in silico anti-inflammatory findings
Kandil et al. [119]	IV Mg (2 g), metoclopramide (10 mg), or prochlorperazine (10 mg)	Single IV dose	157 adult ED migraine patients, randomized into three groups	NRS scores at 30, 60, and 120 min; ED stays; rescue meds; adverse events	No significant difference at 30 or 60 min (e.g., ΔNRS score at 60 min: Mg: –4, *p* = 0.27); similar side effect rates
Gaul et al. [120]	Oral Mg (1100 mg) + riboflavin (400 mg) + CoQ10 (150 mg) (combined supplement)	3 months (after the 4-week baseline)	130 adults with ≥3 migraines/month	Migraine days, pain intensity, HIT-6 scores, subjective benefit	Days: –1.8 vs. –1.0 (NS); pain (*p* = 0.03), HIT-6 scores (*p* = 0.01), subjective efficacy (*p* = 0.01)
Rahimdel et al. [121]	IV MgSO_4_ (1 g in 100 mL saline) vs. DHE	Single IV dose	120 severe migraine patients in the ER	VAS scores at 30, 60, and 90 min	Mg group significantly better at 60 and 90 min (VAS scores: 2.48 vs. 3.48 at 90 min; *p* < 0.05)
Ginder et al. [122]	IV MgSO_4_ (1 g) vs. prochlorperazine	Single IV dose	36 ED patients with acute headache	VAS score before and 30 min post-infusion	Pain relief: 90% (prochlorperazine) vs. 56% (Mg), significant; Mg’s effect not related to serum Mg levels
Khani et al. [123]	Oral magnesium (500 mg/day), sodium valproate (400 mg/day), and their combination for 3 months	A: VPA 200 mg BID + P; B: VPA 200 mg BID + Mg 250 mg BID; C: Mg 250 mg BID + P	222 patients (18–65 years), ≥4 migraines/month; three randomized groups	Frequency, severity, duration, painkillers/month, MIDAS scores, HIT-6 scores	All groups improved (*p* < 0.001); combo > valproate > Mg alone; greater MIDAS/HIT-6 score reduction in combo and valproate groups (*p* < 0.001)
Karimi et al. [124]	Oral magnesium oxide (500 mg BID) vs. sodium valproate (400 mg BID)	8 weeks, crossover	70 migraine patients; 63 completed	Monthly attack frequency, headache days, headache hours	No significant difference between treatments; both effective and safe
Tarighat Esfanjani et al. [125]	Oral magnesium oxide (500 mg/day) and L-carnitine (500 mg/day)	12 weeks; Mg 500 mg/day, L-carnitine 500 mg/day, combo = same doses	133 migraine patients, randomized into three intervention groups and one control group	Attacks/month, days/month, severity, and serum Mg and L-carnitine levels	All interventions reduced migraine indicators (*p* < 0.05); ANOVA: significant frequency reduction (*p* = 0.008); Mg had an independent significant effect
Köseoglu et al. [126]	Oral Mg citrate (600 mg/day)	3 months	40 patients with migraine without aura (30 Mg, 10 placebo), aged 20–55	Attack frequency, severity, P1 amplitude (VEP), cortical perfusion (SPECT)	Mg group: ↓ frequency (*p* = 0.005), ↓ severity (*p* < 0.001), ↓ P1 (*p* < 0.05); ↑ cortical perfusion (*p* = 0.001–0.01); all vs. placebo significant

Abbreviations: BID = twice daily; Mg/MgSO_4_: magnesium/magnesium sulfate; IV: intravenous; VAS: Visual Analogue Scale; NRS: Numeric Rating Scale; NNH: Number Needed to Harm; NNT: Number Needed to Treat; MIDAS: Migraine Disability Assessment Scale; HIT-6: Headache Impact Test; CGRP: Calcitonin-Gene-Related Peptide; HIIT: high-intensity interval training; CoQ10: Coenzyme Q10; DHE: Dihydroergotamine; VEP: Visual Evoked Potential; SPECT: Single-Photon Emission Computed Tomography; ↓: decrease, ↑ increase.

**Table 5 nutrients-17-02216-t005:** A summary of the studies investigating the role of magnesium and the diet in dementia and cognitive decline.

Study (Ref.)	Intervention	Duration	Participants	Outcomes Measured	Key Findings	Cognitive Outcome Measures
Alam et al. [155]	Baseline serum magnesium levels	24 years	12,040 dementia-free adults	Dementia incidence; cognitive function	Lowest serum magnesium quintile linked to a 24% higher dementia risk (HR = 1.24); no link with cognitive decline rates	Dementia incidence, DWRT, DSST, WFT
Zhu et al. [151]	Oral personalized magnesium supplementation vs. a placebo	12 weeks	250 (subgroup >65 years, high Ca:Mg ratios)	Cognition; APOE gene methylation	Reducing Ca:Mg to ~2.3 improved cognition by 9.1% (*p* = 0.03), mediated partly by epigenetic changes	MoCA
Ni et al. [159]	Oral Mg-L-threonate (1.2 g/day) vs. a placebo	12 weeks; 3- and 6-month follow-up	109 post-breast cancer surgery patients	Pain, mood, sleep, cognition	No significant benefits in terms of pain, mood, sleep, or cognition; combination therapies suggested	TICS
Ozawa et al. [160]	Dietary intake of K, Ca, and Mg (FFQ)	17 years of follow-up	1081 Japanese adults ≥60, dementia-free	Incidence of all-cause dementia, VaD, AD	Higher intake of K, Ca, and Mg linked to a lower risk of all-cause dementia and vascular dementia; no link with AD	Incidence of dementia, AD, and VaD
Cherbuin et al. [161]	Dietary intake of Mg, K, Fe (questionnaire)	8 years of follow-up	1406 cognitively healthy adults (mean age: 62.5)	Risk of MCI and mild cognitive disorders	Higher Mg intake associated with reduced MCI/MCD risk; higher K and Fe intake linked to increased risk	MMSE
Lo et al. [162]	Dietary and supplemental Mg intake (FFQ)	>20 years of follow-up	6473 postmenopausal women (65–79 years)	Physician-adjudicated MCI and probable dementia (PD)	Moderate Mg intake (Q2–Q5) associated with a lower risk of MCI and PD; non-linear relationship, no significant effect at extremes	Modified MMSE
Kimura et al. [163]	Dietary intake of vegetables, fruits, and nutrients (FFQ)	24 years	1071 Japanese adults ≥60, dementia-free	Incident dementia, AD, VaD	Higher vegetable intake linked to a 27% lower dementia risk and a 31% lower AD risk; no association with VaD; higher Mg, Ca, K, vitamin A and C, and riboflavin intake also protective; fruit intake not significant	Incidence of dementia, AD, and VaD
Tao et al. [164]	Total magnesium intake (diet + supplements)	NHANES 2011–2014	2508 adults ≥60 years	Global cognitive z-scores; serum vitamin D levels	Higher magnesium intake linked to better cognition, especially in women, non-Hispanic Whites, and those with sufficient vitamin D levels	CERAD, AF, DSST
Tao et al. [165]	Dietary magnesium intake (FFQ)	6 years	5663 adults ≥55 years	Cognitive tests (MMSE, DSST, CDT); impairment risk	Higher intake linked to a lower cognitive impairment risk in men (MMSE, DSST); in women, only MMSE results were significant. This effect was independent of vitamin D in men.	MMSE, DSST, CDT
Luo et al. [166]	Dietary Ca, Mg, and Ca:Mg ratio (FFQ)	5 years	1565 dementia-free urban older adults	Incident dementia (DSM-IV)	Lowest tertile for Ca (<339.1 mg/day) and Mg (<202.1 mg/day) intake linked to the highest dementia risk; in the subgroup with Ca:Mg ≤1.69, a Mg intake >267.5 mg/day increased dementia risk (HR: 3.97)—highlights the importance of Ca:Mg balance	MMSE, Conflicting Instructions Task, Stick Design Test, modified Common Objects Sorting Test, Auditory Verbal Learning Test, modified Fuld Object Memory Evaluation, and Trail Making Test Parts A and B
Tzeng et al. [167]	Oral magnesium oxide (MgO) use vs. no use	10 years	1547 MgO users vs. 4641 matched controls (≥50 years)	Incidence of dementia (Cox regression)	MgO users showed a significantly lower dementia risk (adjusted HR: 0.517, *p* = 0.001)	Incidence of AD, VaD, and non-VaD
Zhang et al. [168]	Oral magnesium L-threonate (Magtein^®^ PS: 400 mg Mg L-threonate + phosphatidylserine + vitamins C, D, B6)	30 days	109 healthy Chinese adults (18–65 years)	Clinical Memory Test (5 subtests + memory quotient	The Magtein^®^PS group showed significant improvements in all memory domains; older adults benefited most.	The Clinical Memory Test
Takeuchi et al. [169]	Web-based 24 h dietary assessment of macronutrients and minerals (alcohol, sugars, fats, magnesium, protein)	12 years	161,376 middle-aged and older UK adults (the UK Biobank cohort)	Incidence of all-cause dementia (hospital records, death registry)	Higher dementia risk associated with no alcohol intake and high sugar/carbohydrate, very low/high fat, very low/high Mg, and the highest protein intake; moderate intake linked to lower risk	Dementia incidence
Cohen-Hagai et al. [170]	Oral magnesium oxide (520 mg) vs. a placebo	8 weeks	29 outpatients with liver cirrhosis	Serum/intracellular Mg; cognition	83% had cognitive impairments; cognitive scores correlated with Mg levels	The MoCA, the CCT, digit span examinations, and the Lowenstein Occupational Cognitive Assessment
Alateeq et al. [171]	Dietary magnesium intake (24 h recall); latent trajectory analysis	17 years	6001 individuals aged 40–73	Brain volumes (GM, WM, hippocampus), white matter lesions (WMLs), blood pressure	Higher Mg intake associated with larger brain volumes and fewer WMLs, especially in post-menopausal women; BP did not mediate outcomes	Brain volumes (GM, WM, LHC, RHC, WMLs)

Abbreviations: Mg: Magnesium; Ca: Calcium; K: Potassium; Fe: Iron; FFQ: Food Frequency Questionnaire; APOE: Apolipoprotein E Gene; MHE: Minimal Hepatic Encephalopathy; MCI: Mild Cognitive Impairment; MCD: Mild Cognitive Disorder; PD: Probable Dementia; AD: Alzheimer’s Disease; VaD: Vascular Dementia; HR: Hazard Ratio; MgO: Magnesium Oxide; GM: Gray Matter; WM: White Matter; WMLs: White Matter Lesions; BP: Blood Pressure; DSM-IV: Diagnostic and Statistical Manual of Mental Disorders, Fourth Edition; GAD-7: Generalized Anxiety Disorder 7-Item Scale; PHQ-9: Patient Health Questionnaire, 9-Item (Depression Scale); PSQI: Pittsburgh Sleep Quality Index; SF-MPQ: Short-Form McGill Pain Questionnaire; TICS: Telephone Interview for Cognitive Status; MMSE: Mini-Mental State Examination; DSST: Digit Symbol Substitution Test; DWRT: Delayed Word Recall Test; WFT: Word Fluency Test; MoCA: Montreal Cognitive Assessment; CERAD: Consortium to Establish a Registry for Alzheimer’s Disease; AF: Animal Fluency; CDT: Clock Drawing Test; CCT: Clock Completion Test.

## Data Availability

Data sharing is not applicable to this article, as no new data were created or analyzed in this study.

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
