# Peer review of "The Role of Magnesium in Depression, Migraine, Alzheimer’s Disease, and Cognitive Health: A Comprehensive Review"

_nutrients, 2025, doi:10.3390/nu17132216_

Round 1
Reviewer 1 Report
Comments and Suggestions for Authors
The Review by Peter Varga and colleagues deals with an interesting topic: the role of magnesium in the modulation of several neurological disorders, such as depression, migraine, Alzheimer's disease and cognitive decline in general. The manuscript tries to explain how magnesium can modulate these pathological processes, describing the possible mechanisms of action, and reports the results of several clinical trials. Overall the manuscript is quite complete and well written, even if there are some aspects that could be improved.
- Some sentences are unclear:
line 156 - Was Mg used as the only treatment?
line 161-162 - Did Mg improve depressive symptoms compared to placebo? (not vs imipramine)
line 168 - Who is the control group? Only vitamin D?
line 178 - What is meant by "prudent" diet?
line 180-181 - It is unclear: did depression improve or not?
line 239-240 - Is the difference significant or not?
line 244-245 - Compared to what?
- Horizontal and vertical lines would be useful in all tables to read the text more clearly; it is often difficult to understand what it refers to, especially when there are long sentences.
- In figure 1 the vertical writings in the boxes on the left are not legible, the text is cut off.
- most important aspect: in all the clinical trails reported it would be useful to know (if the data is reported in the cited papers) the initial, pre-treatment serum Mg level, not only the final one. This would allow us to understand if there is a correlation between serum Mg levels and the pathological state. This could also explain the variability observed in the results of the different trails: it depends on the starting point (patient with low Mg levels or normal?)
- some papers listed in the references are incomplete (e.g. ref. 1: journal? year? ref. 92: the title seems incomplete) or not very relevant (ref. 34, 38, 40, 55, 65, 91).
Some sentences can be rewritten to better clarify the concept
Author Response
The Review by Peter Varga and colleagues deals with an interesting topic: the role of magnesium in the modulation of several neurological disorders, such as depression, migraine, Alzheimer's disease and cognitive decline in general. The manuscript tries to explain how magnesium can modulate these pathological processes, describing the possible mechanisms of action, and reports the results of several clinical trials. Overall the manuscript is quite complete and well written, even if there are some aspects that could be improved.
Response: Thank you for the overall positive feedback.
Some sentences are unclear:
line 156 - Was Mg used as the only treatment?
line 161-162 - Did Mg improve depressive symptoms compared to placebo? (not vs imipramine)
line 168 - Who is the control group? Only vitamin D?
line 178 - What is meant by "prudent" diet?
line 180-181 - It is unclear: did depression improve or not?
line 239-240 - Is the difference significant or not?
line 244-245 - Compared to what?
Response: Thank you for pointing out these ambiguities. We have reviewed each of the noted sentences and revised them for greater clarity. Specifically, we clarified whether magnesium was used as a monotherapy or compared to placebo, defined dietary patterns such as the ‘prudent diet,’ and specified control groups and whether outcomes were statistically significant. These clarifications aim to improve the precision and readability of the manuscript.
Horizontal and vertical lines would be useful in all tables to read the text more clearly; it is often difficult to understand what it refers to, especially when there are long sentences.
Response: Thank you for your suggestion. We agree that additional lines can improve table readability. However, we have followed the formatting guidelines provided by the Nutrients journal. Should the editorial team advise otherwise, we would be happy to adjust the table formatting accordingly.
In figure 1 the vertical writings in the boxes on the left are not legible, the text is cut off.
Response: Thank you for your observation. We corrected Figure 1.
most important aspect: in all the clinical trails reported it would be useful to know (if the data is reported in the cited papers) the initial, pre-treatment serum Mg level, not only the final one. This would allow us to understand if there is a correlation between serum Mg levels and the pathological state. This could also explain the variability observed in the results of the different trails: it depends on the starting point (patient with low Mg levels or normal?)
Response: Thank you for this insightful observation. We agree that baseline (pre-treatment) serum magnesium levels are crucial for understanding both the potential effectiveness of supplementation and inter-study variability. Unfortunately, not all clinical trials reported these initial values, limiting our ability to systematically compare outcomes based on magnesium status at baseline. Where such data were available, we have noted it in the tables and discussion. We have also added a comment in the manuscript emphasizing the need for future trials to consistently report both baseline and post-treatment magnesium levels to improve interpretability.
some papers listed in the references are incomplete (e.g. ref. 1: journal? year? ref. 92: the title seems incomplete) or not very relevant (ref. 34, 38, 40, 55, 65, 91).
Response: Thank you for pointing out these issues.
We have reviewed Ref 1, which is a Chapter from the book Phytochemicals (ISBN: 978-1-536215-478-8).
Ref 92 appears complete and correctly formatted.
Ref 34, 38, 40, 55, 65, 91 has been removed.
Reviewer 2 Report
Comments and Suggestions for Authors
Ref.: nutrients-3722729
This is a comprehensive review on the role of magnesium in certain neurological disorders,
which is increasingly recognized recently.
The authors describe the pathophysiological and biochemical role of magnesium in
depression, migraine and cognitive disorders including Alzheimer’s disease. Evidence from
biochemical, clinical and epidemiological studies, the neuroprotective and anti-inflammatory
effects and the therapeutic and/or preventive potential in the above disorders, all are included
and extensively reviewed.
The review is very informative and educative, data are appropriately summarized in tabular
form and references are up-to-date.
One minor point: I believe that this paper is a review and not a meta-analysis. Thus in pages
32 and 92, please change “meta-analysis” to “review”
Author Response
Thank you for the overall positive evaluation! We have changed the terms in the respective sections to “review.”
Reviewer 3 Report
Comments and Suggestions for Authors
Comments to the Authors
In addition, the table 1 select patients with more than 18 years old. However, cognitive impairment are present at later stages (>65 years old). Thus, comparisions between older and young patients could affect the results and it is neccesary focus on similar range of ages for real comparisons between neurodegenerative markers. The magensium dosage and administration route is other factor that should be discussed in detail I would suggest include a brief description or link about all evaluated items for depression scales (DASP-42, Becks, etc) as supplementary material. Many of induced-Mg++ reported antidepressant effects are not consistent with changes on systemic markers. For example, the reference 102 confirmed signifficant effects on Beck scale without affecting BDNF levels in supplemented patients with Mg ++ dosage of 500 mg/day. However, 23 elderly patients with DM-II and hypomagnesemia improved higher MgCl2 levels when they wee supplemented with 450 mg/day for 12 weeks without affecting depressive scores. In addition, the degree of depression is not weel defined in some evaluated papers. For example, they showed the reference 104 that evaluate 60 long-Covid 19 patients with mild-to-moderate depression but it is not clear if this consideration were tested in many papers from table-3. Additionally,. Vitamine D combination with Magnesiumreduced inflammatory markers without affecting BDI-II scores. As a whole, these studies mixture diabetes, combined treatments with vitamin D, SSRI drugs but the inclussion criteria is confuse again. In fact, some comorbilities have been also added without a proper debate in the discussion section.
The role of magnesium in the pathophysiology of migraine has been properly justify but its relevance to treat Alzheimer disease is not clear for me. I have seen a p=0.59 value as p-trend in table3 , reference 111. Maybe, they wanted to say p=0.059. In this case, correct this value in table3. In general, discuss the relevance of mangesium form supplementation and routes of administration in the discussion. The table-4 also indicate results of combined Mg++ plus metoclopramide treatments, including more published trials in this table 4. I would suggest include different tables for Magnesium supplementation alone and other for combined treatments with phytonaturals (Q10, vitamin D, B6) or SSRI antidepressant drugs plus magnesium supplementation, including the exaclty route of administration for the treatment of migraine. I would remove the cyte 121 given the low size sample (n=15) in this table-4. I also recommend you describe key findings without addding p values. It is better describe the relevant physiological findings by Mg++ treatment in this table.
The cyte 160 stablishs a relathionship between magnesium and diabetes and Alzheimer but this evidence is not supported by 103. Are these discrepances associated to different size samples in both studies?
The line 381 indicate the improvement of 9.1 % on cognitive disfunction, which is partially mediated by epigenetic changes. How do you justify this low porcentage as improvement in cognitive function? Is this relevant from a clinical view point? What do you mean cognitive disfucntion for you? What does mean the Ca++: Mg++ ratio alteration from a pathophysiological view point?
The table-5 shows a summunary of studies investigating the role of magnesium and diet against dementia and cognitive decline. However, cytes 161 162 163 showed results that did not affect cognition while references 168 169 and 172 described effects on cognition associated with combined treatments with diet/supplements without a clear study design. Shall you indicate the marker able to evaluate cognition among these comparative studies from table-5?
Magnesium binds to NMDA receptor but does not interfere with serotonine or GABA receptor. How you explain that magnesium could alter GABA-ergic and serotoninergic signaling if does not bind to these receptors? Maybe, I don`t understantd you. Thanks¡
The discussion is obvious and general. However, it is necessary include a discussion about the comparison between findings showed in tables For example, lines 475-478 from the discussion belong to the introduction. The role of magnesium dosage, route administration and time are factors that should be included within the discussion section. Please, add also adverse or null effects of magnesium in Alzheimer disease pathology.
How is possible to justify induced-antidepressant effects of magnesium in some published studies in your tables? Is NMDA receptor function affected=? Shall you stablish a relationship between magnesium treatment and cognitive improvements? Shall you indicate the optimal dosage for Magnesium published in the literature?
The role of magnesium as preventive agent against Alzheimer disease has been stablished. However, there are also evidences on a detrimental role of magnesium in certain neurological diseases, including Alzheimer when its levels are very high. Please, shall you include evidence also in this review. These authors have measured magnesium levels in plasma but it is necessary include more evidence on detrimental or lack of beneficial effects by magnesium supplementation in patients.
My Decision is Major revision
Comments on the Quality of English LanguageThe English Language can be improve it.
Author Response
In addition, the table 1 select patients with more than 18 years old. However, cognitive impairment are present at later stages (>65 years old). Thus, comparisions between older and young patients could affect the results and it is neccesary focus on similar range of ages for real comparisons between neurodegenerative markers. The magensium dosage and administration route is other factor that should be discussed in detail I would suggest include a brief description or link about all evaluated items for depression scales (DASP-42, Becks, etc) as supplementary material. Many of induced-Mg++ reported antidepressant effects are not consistent with changes on systemic markers. For example, the reference 102 confirmed signifficant effects on Beck scale without affecting BDNF levels in supplemented patients with Mg ++ dosage of 500 mg/day. However, 23 elderly patients with DM-II and hypomagnesemia improved higher MgCl2 levels when they wee supplemented with 450 mg/day for 12 weeks without affecting depressive scores. In addition, the degree of depression is not weel defined in some evaluated papers. For example, they showed the reference 104 that evaluate 60 long-Covid 19 patients with mild-to-moderate depression but it is not clear if this consideration were tested in many papers from table-3. Additionally,. Vitamine D combination with Magnesiumreduced inflammatory markers without affecting BDI-II scores. As a whole, these studies mixture diabetes, combined treatments with vitamin D, SSRI drugs but the inclussion criteria is confuse again. In fact, some comorbilities have been also added without a proper debate in the discussion section.
Response: Thank you for this insightful comment. We agree that cognitive impairment is more prevalent in individuals over 65 years. In our review, we emphasized findings from studies specifically targeting older adults where applicable (e.g., dementia-focused trials). However, as this is a narrative review, we included studies covering the adult lifespan to provide a broader overview. Magnesium dosage and administrative routes can also contribute to the diversity of findings. They are all indicated in the tables (Intervention column). We have clarified these points in the Limitations.
Thank you for the suggestion for the supplementary material. We added a new file as a supplement with the short descriptions of the different psychometric inventories mentioned in the review.
The results of included articles on depression indeed varied greatly, which can be contributed to many factors. like sample sizes, measurement methods, and magnesium supplementation protocols, making it difficult to compare results and draw general conclusions.
The role of magnesium in the pathophysiology of migraine has been properly justify but its relevance to treat Alzheimer disease is not clear for me. I have seen a p=0.59 value as p-trend in table3 , reference 111. Maybe, they wanted to say p=0.059. In this case, correct this value in table3. In general, discuss the relevance of mangesium form supplementation and routes of administration in the discussion. The table-4 also indicate results of combined Mg++ plus metoclopramide treatments, including more published trials in this table 4. I would suggest include different tables for Magnesium supplementation alone and other for combined treatments with phytonaturals (Q10, vitamin D, B6) or SSRI antidepressant drugs plus magnesium supplementation, including the exaclty route of administration for the treatment of migraine. I would remove the cyte 121 given the low size sample (n=15) in this table-4. I also recommend you describe key findings without addding p values. It is better describe the relevant physiological findings by Mg++ treatment in this table.
Response: Thank you for your comment! Upon revisiting the original source of reference 111, we confirm that the correct value is p = 0.59. The ‘Discussion’ section has also been revised about the relevance of magnesium supplementation form and routes of administration. We really appreciate your idea of dividing Table 4 based on combination of magnesium supplementation. Table 4 was amended accordingly. While we recognize that the small sample size in citation 121 (n = 15) limits the statistical power of the findings, we chose to retain it due to its noteworthy clinical results, particularly the high rate of symptom resolution. Rather than omitting the study, we now clearly indicate in the text that should be interpreted with caution. Based on Reviewer 3 comment on p-values, we decided not to remove them to keep all relevant information in one table.
The cyte 160 stablishs a relathionship between magnesium and diabetes and Alzheimer but this evidence is not supported by 103. Are these discrepances associated to different size samples in both studies?
Response: Thank you for pointing this out. Yes, part of the discrepancy likely stems from differences in sample size and study design. Reference 160 is a large cohort study with mediation analysis, while reference 103 is a small RCT with a specific subpopulation (elderly with diabetes).
The line 381 indicate the improvement of 9.1 % on cognitive disfunction, which is partially mediated by epigenetic changes. How do you justify this low porcentage as improvement in cognitive function? Is this relevant from a clinical view point? What do you mean cognitive disfucntion for you? What does mean the Ca++: Mg++ ratio alteration from a pathophysiological view point?
Response: We clarified in the revised manuscript that the 9.1% change refers to relative improvement in cognitive scores, which—although modest—was statistically significant and potentially meaningful in early cognitive decline. We also elaborated on the Ca:Mg ratio’s biological relevance.
The table-5 shows a summunary of studies investigating the role of magnesium and diet against dementia and cognitive decline. However, cytes 161 162 163 showed results that did not affect cognition while references 168 169 and 172 described effects on cognition associated with combined treatments with diet/supplements without a clear study design. Shall you indicate the marker able to evaluate cognition among these comparative studies from table-5?
Response: Thank you for this important observation. We have now added a column to Table 5 specifying the cognitive outcome measures used in each study.
Magnesium binds to NMDA receptor but does not interfere with serotonine or GABA receptor. How you explain that magnesium could alter GABA-ergic and serotoninergic signaling if does not bind to these receptors? Maybe, I don`t understantd you. Thanks¡
Response: Thank you for your question. Magnesium does not bind directly to GABA or serotonin receptors but modulates their signaling indirectly through effects on receptor sensitivity, synaptic plasticity, and ion channel regulation.
The discussion is obvious and general. However, it is necessary include a discussion about the comparison between findings showed in tables For example, lines 475-478 from the discussion belong to the introduction. The role of magnesium dosage, route administration and time are factors that should be included within the discussion section. Please, add also adverse or null effects of magnesium in Alzheimer disease pathology.
Response: Thank you for your feedback. We amended the ‘Discussion’ accordingly.
How is possible to justify induced-antidepressant effects of magnesium in some published studies in your tables? Is NMDA receptor function affected=? Shall you stablish a relationship between magnesium treatment and cognitive improvements? Shall you indicate the optimal dosage for Magnesium published in the literature?
Response: Thank you for your thoughtful comment. We agree that the mechanisms underlying magnesium’s antidepressant effects, particularly its modulation of NMDA receptor activity, are important and well-established in the literature. These pathways have been briefly addressed in the current version of the manuscript to maintain focus and scope, as our primary aim was to provide a comprehensive overview rather than an in-depth mechanistic review. Regarding dosage, we have summarized ranges reported in the cited studies, but acknowledge that further research is needed to establish optimal dosing. We believe the current level of detail provides sufficient context for the purpose and structure of this narrative review.
The role of magnesium as preventive agent against Alzheimer disease has been stablished. However, there are also evidences on a detrimental role of magnesium in certain neurological diseases, including Alzheimer when its levels are very high. Please, shall you include evidence also in this review. These authors have measured magnesium levels in plasma but it is necessary include more evidence on detrimental or lack of beneficial effects by magnesium supplementation in patients.
Response: Thank you for your comment. We added discussion on the detrimental effects of both hypomagnesemia and hypermagnesemia, including epidemiological studies suggesting a U-shaped curve in dementia risk. We also emphasized the importance of maintaining magnesium within an optimal serum range (0.80–0.89 mmol/L).
Supplementary material
- Psychometric Inventories in Included Studies:
- Beck Depression Inventory (BDI): a widely used, self-report inventory. It consists of 21 items, each scored 0-3, total score range is 0-63. It is designed for screening (not diagnostics) with high internal consistency, high sensitivity (100%) and high specificity (96%) [1].
- BDI-II: The second version of BDI designed for older adults. Items were aligned with DSM-IV criteria [1].
- Depression Anxiety and Stress Scale (DASS-42): a 42-item self-report questionnaire designed to assess the negative emotional states of depression, anxiety, and stress. The primary purpose of the DASS is to determine the severity of core symptoms related to depression, anxiety, and stress [2].
- Generalized Anxiety Disorder 7-item (GAD-7): a self-report screening tool for generalized anxiety disorder. The items focus on frequency of anxiety symptoms over the past two weeks. Score range is 0-21 [3].
- Hamilton Depression Rating Scale (HDRS): a clinician-administered questionnaire commonly used to evaluate the severity of depression in individuals already diagnosed with a depressive disorder. The 17-item version is used most commonly, score range is 0-52 [4].
- Patient Health Questionnaire-9 (PHQ-9): a widely used, self-administered screening tool designed to assess the severity of depressive symptoms over the preceding two weeks. It is based on the nine diagnostic criteria for major depressive disorder from the DSM-IV and serves both as a screening instrument and a monitoring tool for treatment response. Score range is 0-27 [4].
References
- Edelstein, B.A.; Drozdick, L.W.; Ciliberti, C.M. Chapter 1 - Assessment of Depression and Bereavement in Older Adults. In Handbook of Assessment in Clinical Gerontology (Second Edition), Lichtenberg, P.A., Ed.; Academic Press: San Diego, 2010; pp. 3–43.
- Lovibond, P.F.; Lovibond, S.H. The Structure of Negative Emotional States - Comparison of the Depression Anxiety Stress Scales (Dass) with the Beck Depression and Anxiety Inventories. Behav Res Ther 1995, 33, 335–343, doi:Doi 10.1016/0005-7967(94)00075-U.
- Spitzer, R.L.; Kroenke, K.; Williams, J.B.W.; Löwe, B. A brief measure for assessing generalized anxiety disorder -: The GAD-7. Arch Intern Med 2006, 166, 1092–1097, doi:DOI 10.1001/archinte.166.10.1092.
- Ma, S.; Yang, J.; Yang, B.; Kang, L.; Wang, P.; Zhang, N.; Wang, W.; Zong, X.; Wang, Y.; Bai, H.; et al. The Patient Health Questionnaire-9 vs. the Hamilton Rating Scale for Depression in Assessing Major Depressive Disorder. Front Psychiatry 2021, 12, 747139, doi:10.3389/fpsyt.2021.747139.
Reviewer 4 Report
Comments and Suggestions for Authors
Although the review is thorough, it is devoid of a distinct hypothesis or original addition to the subject. The manuscript provides no conceptual framework or new insights, only a generic synthesis of previous research. In the opening, make clear the review's unique perspective. For example, is this the first time magnesium has been compared across these three illnesses, or are you suggesting a common molecular pathway?
Despite mentioning the use of several datasets and a PICO framework, the authors' systematic methodology is not sufficiently explained. Provide a detailed description of the research selection procedure, risk of bias assessment, and data synthesis criteria, along with a PRISMA flow diagram.
While some studies are statistically detailed in their summaries, others are not. Confidence intervals and p-values are used inconsistently in the review. Ensure that results are reported consistently in both text and tables. If at all possible, include p-values, effect sizes, and confidence intervals.
Many inferences about causality are made from observational studies without enough hesitancy. Make a clear distinction between interventional and correlational data, and adjust conclusions appropriately.
Magnesium's function in inflammation, oxidative stress, and neurotransmission is repeatedly described throughout the long manuscript. Reduce the length of the narrative by cutting out unnecessary parts and using infographics or summary tables.
Contradictory data are seldom properly evaluated in the discussion, particularly in studies pertaining to depression and migraines. Incorporate a comparative analysis of research with varying findings, possible causes of disparities, and the constraints of the existing body of information.
The manuscript lacks visual aids and is primarily text-based. Include one or two mechanistic figures that summarize the ways in which magnesium influences oxidative stress, neurotransmission, and brain inflammation in each of the three illnesses.
There are sudden changes between the sections on dementia, migraine, and depression. Use more concise subheadings with a standardized format for every condition (e.g., pathophysiology, evidence, limitations, clinical recommendations) and include a roadmap at the conclusion of the introduction.
The constraints section is shallow, mentioning broad issues like heterogeneity. Talk on the particular drawbacks of the studies that were cited, such as their small sample sizes, low magnesium bioavailability, and brief intervention times.
There is a lack of clarity on the implications for clinical practice. Include a specific part on Clinical Relevance or Future Perspectives that summarizes useful recommendations and areas that require clinical studies.
Author Response
Although the review is thorough, it is devoid of a distinct hypothesis or original addition to the subject. The manuscript provides no conceptual framework or new insights, only a generic synthesis of previous research. In the opening, make clear the review's unique perspective. For example, is this the first time magnesium has been compared across these three illnesses, or are you suggesting a common molecular pathway?
Response: Thank you for pointing this out. In the revised Introduction, we now explicitly state that this is the first narrative review to compare the role of magnesium across three major neurological disorders—depression, migraine, and Alzheimer’s disease—with an emphasis on shared mechanisms and clinical implications.
Despite mentioning the use of several datasets and a PICO framework, the authors' systematic methodology is not sufficiently explained. Provide a detailed description of the research selection procedure, risk of bias assessment, and data synthesis criteria, along with a PRISMA flow diagram.
Response: Thank you for comment! Since this is a narrative review and not a systematic review, we did not apply PRISMA.
While some studies are statistically detailed in their summaries, others are not. Confidence intervals and p-values are used inconsistently in the review. Ensure that results are reported consistently in both text and tables. If at all possible, include p-values, effect sizes, and confidence intervals.
Response: Thank you for pointing this out. We have reviewed the tables and text to ensure greater consistency in the reporting of statistical outcomes.
Many inferences about causality are made from observational studies without enough hesitancy. Make a clear distinction between interventional and correlational data, and adjust conclusions appropriately.
Response: Thank you. We revised multiple statements in the text to emphasize associative (rather than causal) relationships where observational studies were cited, and we clearly distinguish between RCTs and cohort findings.
Magnesium's function in inflammation, oxidative stress, and neurotransmission is repeatedly described throughout the long manuscript. Reduce the length of the narrative by cutting out unnecessary parts and using infographics or summary tables.
Response: Thank you for this helpful suggestion. We reviewed the manuscript and reduced redundancy by condensing overlapping mechanistic explanations across sections.
Contradictory data are seldom properly evaluated in the discussion, particularly in studies pertaining to depression and migraines. Incorporate a comparative analysis of research with varying findings, possible causes of disparities, and the constraints of the existing body of information.
Response: Thank you for this important comment. In response, we revised the manuscript to better reflect on contradictory data. We also emphasize the need for standardized protocols to clarify magnesium’s role and improve comparability across future studies.
The manuscript lacks visual aids and is primarily text-based. Include one or two mechanistic figures that summarize the ways in which magnesium influences oxidative stress, neurotransmission, and brain inflammation in each of the three illnesses.
Response: Thank you for this helpful suggestion. We agree that visual representations can enhance reader engagement and understanding. However, in order to maintain the narrative flow and focus on text-based synthesis, we chose not to include mechanistic figures in this version. Instead, we aimed to provide clear and structured descriptions of the shared and distinct mechanisms by which magnesium may influence depression, migraine, and Alzheimer’s disease. We hope the current format still provides sufficient clarity for readers interested in the mechanistic aspects.
There are sudden changes between the sections on dementia, migraine, and depression. Use more concise subheadings with a standardized format for every condition (e.g., pathophysiology, evidence, limitations, clinical recommendations) and include a roadmap at the conclusion of the introduction.
Response: Thank you for this valuable suggestion. To improve coherence and reader navigation, we added a roadmap paragraph at the end of the Introduction to clarify the structure of the review and outline how the conditions will be comparatively discussed in relation to magnesium.
The constraints section is shallow, mentioning broad issues like heterogeneity. Talk on the particular drawbacks of the studies that were cited, such as their small sample sizes, low magnesium bioavailability, and brief intervention times.
Response: Thank you for this constructive comment. We have revised the Limitations section to include more specific concerns related to the cited studies, such as small sample sizes, limited intervention durations, and the use of low-bioavailability magnesium formulations (e.g., magnesium oxide). These limitations may have contributed to the inconsistent findings across studies and reduce the generalizability of the results.
There is a lack of clarity on the implications for clinical practice. Include a specific part on Clinical Relevance or Future Perspectives that summarizes useful recommendations and areas that require clinical studies.
Response: Thank you for your thoughtful suggestion. We agree that highlighting the clinical relevance and future implications of magnesium research is important. We would like to note that these aspects are already addressed in several parts of the manuscript. Specifically, we discuss translational challenges and potential clinical applications of magnesium in the Discussion and within the sections on each condition. We also outline directions for future research, including the need for standardized protocols, long-term studies, and personalized approaches. To improve clarity, we have ensured that these points are clearly articulated and well integrated across the manuscript.
Round 2
Reviewer 3 Report
Comments and Suggestions for Authors
All requirements were solved in this resubmitred version
Comments on the Quality of English LanguagePlease, improve the english style
Reviewer 4 Report
Comments and Suggestions for Authors
The author changed well.